# Glucose fluctuation promotes mitochondrial dysfunctions in the cardiomyocyte cell line HL-1

Patrick Mordel[1], Fanny Fontaine[2], Quentin Dupas[1], Michael Joubert[1,3], Stéphane Allouche[1,2]*

1 Normandie University, UNICAEN, CHU Caen, Signalisation, Electrophysiologie et Imagerie des Lésions d'Ischémie-Reperfusion Myocardique, Caen, France, 2 Department of Biochemistry, CHU de Caen, Caen, France, 3 CHU de Caen, Diabetes Care Unit, Caen, France

* allouche-s@chu-caen.fr

## Abstract

### Aims

Glycemic variability has been suggested as a risk factor for diabetes complications but the precise deleterious mechanisms remain poorly understood. Since mitochondria are the main source of energy in heart and cardiovascular diseases remain the first cause of death in patients with diabetes, the aim of the study was to evaluate the impact of glucose swings on mitochondrial functions in the cardiomyocyte cell line HL-1.

### Methods

HL-1 cells were exposed to low (LG, 2.8 mmol/l), normal (NG, 5.5 mmol/l), high (HG, 25 mmol/l) or intermittent high glucose (IHG, swing between low and high) every 2h during 12h (short-time treatment) or every 12h during 72h (long-time treatment). Anaerobic catabolism of glucose was evaluated by measuring glucose consumption and lactate production, oxidative phosphorylation was evaluated by polarography and ATP measurement, mitochondrial superoxide anions and the mitochondrial membrane potential (MMP) were analysed using fluorescent probes, and the protein oxidation was measured by oxyblot.

### Results

IHG and HG increased glucose consumption and lactate production compared to LG and NG but without any difference between short- and long-time treatments. After 72h and unlike to LG, NG and HG, we didn't observe any increase of the mitochondrial respiration in the presence of succinate upon IHG treatment. IHG, and to a lesser extent HG, promoted a time-dependent decrease of the mitochondrial membrane potential compared to LG and NG treatments. HG and IHG also increased superoxide anion production compared to LG and NG both at 12 and 72h but with a higher increase for IHG at 72h. At last, both HG and IHG stimulated protein oxidation at 72h compared to LG and NG treatments.

**Data Availability Statement:** All relevant data are within the paper and its Supporting Information files.

**Funding:** This work was supported by Université Caen Normandie (Unicaen – France).

**Competing interests:** PM, QD and SA declare that they have no conflict of interest. MJ has received research grants and/or advisory fees from Astrazeneca, Bayer, Boehringer Ingelheim, Lilly and Novonordisk.

## Conclusions

Our results demonstrated that exposure of HL-1 cells to glucose swings promoted time-dependent mitochondrial dysfunctions suggesting a deleterious effect of such condition in patients with diabetes that could contribute to diabetic cardiomyopathy.

## Introduction

The diabetes prevalence all over the world is continuously rising and it's now well demonstrated that this condition is associated with a two to three-fold increase risk of cardiovascular disease (CVD), which remains the primary cause of death in those patients [1]. In patients with type 2 diabetes, reduction of hyperglycemic episodes, reflected by the glycated hemoglobin HbA1c level, significantly decreased death related to diabetes and myocardial infarction [2]. Similar conclusions were obtained in patients with type 1 diabetes treated by intensive diabetes therapies [3]. However, HbA1c only reflects the mean glycemia without considering glycemic variability (GV). GV refers to oscillations of blood glucose concentration (hypo- and hyperglycemia) that occur throughout the day (intra-day variability) but also between days (inter-day variability). A pioneer work reported that, in a population with type 2 diabetes, GV but not HbA1c level, was highly correlated with the urinary 8-iso prostaglandin $F_{2\alpha}$, a stress oxidative marker; those results suggest that GV increases oxidative stress which could contribute to cardiovascular dysfunctions [4]. However, such results were not further confirmed by another study in type 1 diabetes [5]. Although it has not yet been definitively confirmed as an independent risk factor for diabetes complications, many reports have shown that GV was associated with both micro- and macrovascular complications [6]. For exemple, high GV measured early after the onset of a first-episode of acute coronary syndrome is associated with a higher risk of major adverse cardiac and cerebrovascular events at 30 days [7]. All those data suggest a deleterious role of GV on cardiovascular diseases but the cellular and molecular mechanisms remain poorly understood. Very few studies examined the impact of GV on cardiomyocyte [8,9], and to the best of our knowledge, no data on mitochondrial functions are available.

Since mitochondria provide the major part of ATP for the energy needs of cardiomyocytes and diabetes was demonstrated to cause mitochondrial dysfunctions [10], we examined whether permanent high glucose levels or glucose fluctuations would promote changes in glucose metabolism and cause mitochondrial alterations in the cardiomyocyte cell line HL-1. For this purpose, cells were exposed to different glucose concentrations: low glucose (LG, 2.8 mmol/l), normal glucose (NG, 5.5 mmol/l), high glucose (HG, 25 mmol/l) or intermittent high glucose (IHG, swings between low and high) every 2h during 12h (short-time treatment) or every 12h or during 72h (long-time treatment). Then, we examined the glucose catabolism and mitochondrial functions (mitochondrial respiration, ATP production, mitochondrial membrane potential, anion superoxide production and protein oxidation).

## Materials and methods

### Cell culture and treatment

HL-1 mouse atrial cardiomyocyte-derived cells, provided by Dr. William C. Claycomb (Louisiana State University Medical Center, USA), were cultured as previously described [11]. HL-1 cells were cultured either in standard conditions with a normal glucose concentration of 5.5

mmol/l (1 g/l, NG) or with a high glucose concentration of 25 mmol/l (4.5 g/l, HG) for at least 3 weeks.

Cells were subjected to different glucose treatments: 2.8 mmol/l (0.5 g/l, low glucose, LG), 5.5 mmol/l (normal glucose, NG), 25 mmol/l (high glucose, HG) or glucose swing between 2.8 and 25 mmol/l (intermittent high glucose, IHG) every 2h for 12h treatment (short-time treatment) or every 12h for 72h treatment (long-time treatment) (Fig 1).

### Glucose consumption and lactate production

To determine glucose consumption and the lactate production by the HL-1 cells, glucose and lactate concentrations were measured in the culture media at 0h and at different times of glucose treatments (see the arrows in the Fig 1) using the Beckman Coulter AU5800 device.

### Oxygen consumption

After glucose treatments during 12 or 72h, HL-1 cells were harvested and centrifuged for 2 min at 300 g. The oxygen consumption was measured at 37°C on 0.2% (w/v) digitonin-permeabilized cells using a Clark-type oxygen electrode (Oxytherm system, Hansatech Instruments, Cergy Saint Christophe, France). Pellets were resuspended in the respiration buffer (in mmol/l: K-MES 100, $KH_2PO_4$ 5, EGTA 1, EDTA 3, ADP 1 and bovine serum albumin 1 mg/ml, pH 7.4), then cells were placed in the incubation chamber. Several substrates were used: 25 mmol/l succinate + 25 μmol/l rotenone, 80 μmol/l palmitate + 5 mmol/l malate or 10 mmol/l pyruvate + 5 mmol/l malate. Respiration rate was measured using the Oxygraph Plus software, and results were expressed as nmol $O_2.min^{-1}.mg^{-1}$ of protein.

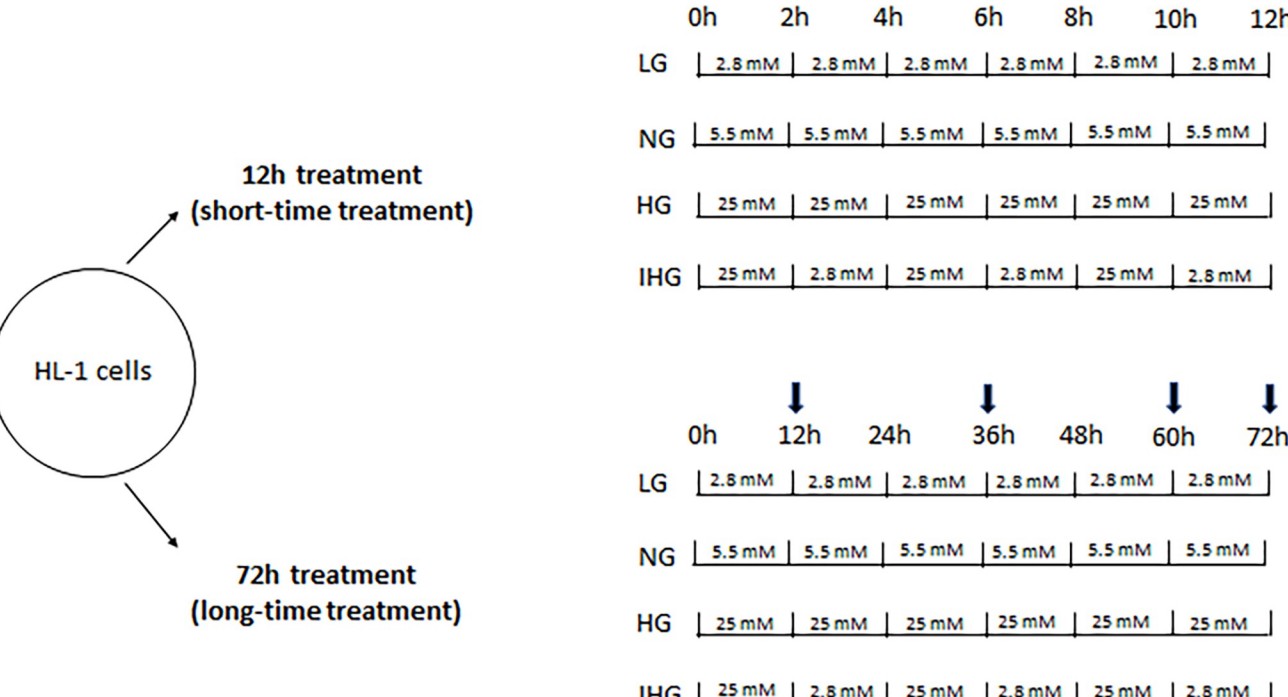

**Fig 1. Schematic representation of short- and long-time treatment of HL-1 cells with different glucose concentrations.** Cells were exposed to low (2.8 mmol/l, LG), normal (5.5 mmol/l, NG), high (25 mmol/l, NG), and intermittent high glucose (25 followed by 2.8 mmol/l, IHG) either during 12 or 72h. The culture medium was changed either every 2h or every 12h for short- and long-time treatment, respectively. Arrows indicate glucose and lactate measurements.

## Assessment of mitochondrial membrane potential (MMP)

Twenty-four hours before experiment, 20,000 cells per well were seeded in 96-well black plate (Greiner bio-one). Cells were exposed to different glucose treatments (LG, NG, HG or IHG) during 12 or 72h. Then, the culture media were replaced by Hank's balanced salt solution (HBSS) (Thermo Fischer) and HL-1 cells were incubated with 200 nmol/l tetramethyl-rhodamine ethyl ester (TMRE, Molecular Probes®) for 30 min at 37˚C. After three washes with HBSS, TMRE fluorescence was measured with a multilabel plate reader (VICTOR™ X4) in triplicate using excitation filter of 544 nm and emission filter of 572 nm.

## Measurement of mitochondrial superoxide anions

The superoxide anion generation was assessed using the mitochondrial fluorescent red probe MitoSOX™ (Molecular Probes®). Twenty-four hours before experiment, 20,000 cells per well were seeded in 96-well black plate. Cells were exposed to different glucose treatments (LG, NG, HG or IHG) during 12 or 72h. At the end of the treatments, the culture media were removed and HL-1 cells were loaded with 5 μmol/l MitoSOX™ in HBSS for 1h at 37˚C. For each treatment condition, cells were incubated in the absence (basal superoxide anion production) or in the presence of 10 μmol/l antimycin A (stimulated-superoxide anion production) for 2h before loading with the fluorescent probe. Cells were washed with HBSS, then fluorescence was measured using a multilabel plate reader (VICTOR™ X4) in triplicate using excitation filter of 485 nm and emission filter of 572 nm.

## Determination of protein oxidation

After glucose treatments, whole cell lysates were prepared as described above using ice-cold lysis RIPA buffer (in mmol/l: Tris-HCl 50, NaCl 150, EGTA 1, IGEPAL 1% (w/v), NaF 0,25% (w/v), and protease inhibitor cocktail, pH 7.5). Total proteins were separated on 12% polyacrylamide gels (12% Mini-PROTEAN® TGX Stain-Free™ Protein Gels, BioRad) and carbonyled proteins content was evaluated according to manufacturer protocol (OxyBlot Protein Oxidation Detection Kit, Merck Millipore). Results were expressed as a ratio of immunoreactive bands and total proteins detected on Stain Free gels.

## Measurement of ATP production

After glucose treatments, HL-1 cells were harvested and centrifuged for 5 min at 100 g, counted and permeabilized using digitonin (20 μg/million) for 5 min. After centrifugation, pellets were resuspended in the respiration buffer (in mmol/l: $KH_2PO_4$ 10, mannitol 300, KCl 10, EGTA 0.5, $MgCl_2$ 5, ADP 0.06, P1,P5-Di(adenosine-5')pentaphosphate 0.075, and bovine serum albumin fatty acid-free 1 mg/ml, pH 7.4), supplemented either with 0.25 mmol/l pyruvate + 0.25 mmol/l malate, 1.25 mmol/l succinate + 0.5 μg/ml rotenone, or no substrate. After 15 min incubation at 37˚C, samples were immediately frozen at -80˚C. ATP concentration was measured using the Luminescent ATP Detection Assay Kit according to manufacturer protocol (Abcam). After subtracting ATP level in the absence of substrate, results for both substrates are expressed in nmol ATP/mg of proteins and adjusted to citrate synthase activity in %.

## Statistical analysis

Data were expressed as mean ± SEM. Two groups comparison was calculated with the Wilcoxon test, multiple comparisons between groups were calculated by one-way repeated ANOVA followed by Dunnett's comparison test or by two-way ANOVA followed by the

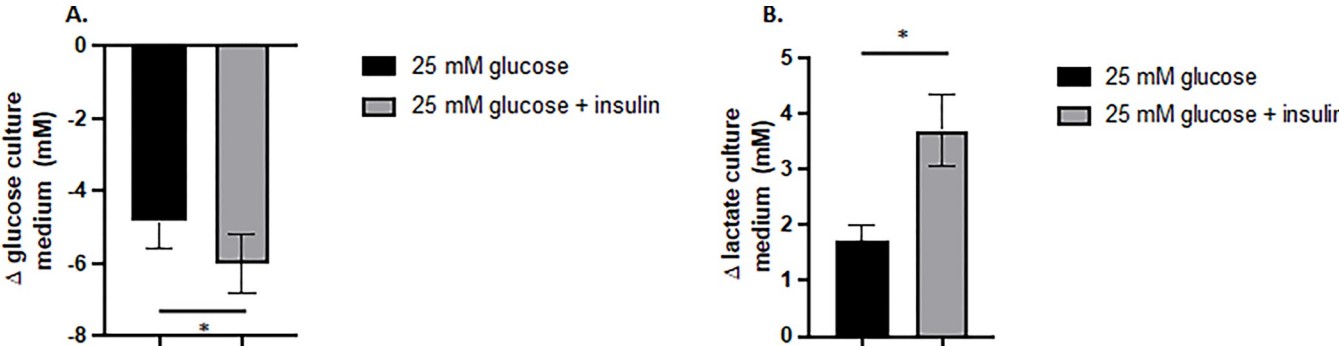

**Fig 2. Effects of insulin on glucose consumption and lactate production.** HL-1 cells were seeded at similar density in T75 flasks, cultured in 25 mmol/l glucose and then exposed or not to 3 IU insulin for 24h. Glucose and lactate concentrations were determined at 0 and 24h in the culture medium to calculate glucose consumption (A) and lactate production (B). Data are the means ± S.E.M of 7 independent experiments. Statistical analysis was conducted using the Wilcoxon test. *, P < 0.05.

Tukey's multiple comparisons test using the GraphPad Prism™ 9 software. Differences were considered significant at P < 0.05.

## Results

### Effect of insulin on glucose consumption and lactate production

HL-1 cells were cultured under standard conditions (25 mmol/l glucose, HG) and we first evaluated the effect of a 24h-treatment with 3 IU insulin on glucose consumption and lactate production. As depicted in the Fig 2A and 2B, we showed that insulin significantly increased glucose consumption (P = 0.031) and lactate production (P = 0.016) by HL-1 cells in the culture medium.

### Effect of glucose treatments on glucose consumption and lactate production

HL-1 cells were cultured in normal glucose medium (NG) for at least 3 weeks then we evaluated glucose consumption and lactate production upon 12h treatment (culture medium changes every 2h, see Fig 1) with low glucose (LG, 2.8 mmol/l), normal glucose (NG, 5.5 mmol/l), high glucose (HG, 25 mmol/l) or intermittent high glucose (IHG, 25 and 2.8 mmol/l). We observed a significant higher glucose consumption for HG and IHG compared to LG or NG at 2 (P = 0.030 HG vs LG; P = 0.014 IHG vs LG), 6 (P < 0.001 IHG vs LG; P = 0.001 IHG vs NG) and 10h (P = 0.003 HG vs LG; P < 0.001 IHG vs LG; P = 0.027 HG vs NG; P < 0.001 IHG vs NG) (S1A Fig). At 12h treatment, we showed a significant difference between HG and IHG (P < 0.001); this is due to low glucose exposure for the last 2h of IHG treatment while for the other times (2, 6 and 10h), cells were exposed to high glucose. We also observed a significant higher lactate production for HG and IHG compared to LG and between HG and NG at 6 (P = 0.002 HG vs LG; P = 0.017 IHG vs LG; P = 0.024 HG vs NG), 10 (P < 0.001 HG vs LG; P = 0.001 IHG vs LG; P = 0.003 HG vs NG), and 12h (P < 0.001 HG vs LG; P = 0.006 HG vs NG) (S1B Fig). At 12h treatment, we also showed a significant difference between HG and IHG (P = 0.021).

When a similar experiment was conducted during 72h (culture medium changes every 12h, see Fig 1), we noticed a significant higher glucose consumption for HG vs LG and NG at 12 (P < 0.001 HG vs LG; P < 0.001 HG vs NG), 60 (P = 0.026 HG vs LG; P = 0.036 HG vs NG), and 72h (P = 0.002 HG vs LG; P = 0.005 HG vs NG) (S1C Fig). HL-1 cells exposed to IHG also

displayed higher glucose consumption vs LG and NG at 12 (P < 0.001 IHG vs LG; P < 0.001 IHG vs NG), 36 (P < 0.001 IHG vs LG; P < 0.001 IHG vs NG) and 60h (P < 0.001 IHG vs LG; P < 0.001 IHG vs NG). As expected, a significant difference was observed between HG and IHG at 72h (P < 0.001 HG vs IHG). Lactate production was higher for the IHG condition at 36 (P = 0.006 IHG vs NG) and 60h (P = 0.009 IHG vs LG; P < 0.001 IHG vs NG) and for the HG condition at 60 (P = 0.022 vs LG; P < 0.001 HG vs NG) and 72h (P < 0.001 HG vs NG) (S1D Fig). Once again, a significant difference for lactate production was observed between HG and IHG at 72h (P = 0.037 HG vs IHG).

We also compared the effect of the short-(12h) and long-time (72h) glucose treatment on glucose consumption and lactate production for the different treatments (LG, NG, HG or IHG) as indicated in the Fig 1. As depicted on the Fig 3A–3D, we found no significant difference on glucose consumption for the different time treatment. In contrast, we observed that lactate production was higher for NG and HG only at the end of the first glucose swing (i.e. 2h for short treatment vs 12h for long treatment) (Fig 3E–3H).

We also checked that when HL-1 cells were cultured under standard conditions (25 mmol/l) or in normal glucose (5.5 mmol/l) medium for at least 3 weeks, there was no impact on glucose consumption (S2 Fig) or on lactate production (S3 Fig) upon 12h treatment with LG, NG, HG or IHG.

## Effect of glucose treatments on oxygen consumption

In a next set of experiments, we examined whether glucose treatments would impact the mitochondrial respiration. Cells were exposed either during 12 or 72h with different treatments (LG, NG, HG or IHG), then oxygen consumption was measured by polarography using three different substrates: succinate, palmitate and pyruvate. At 12h, we observed a higher mitochondrial respiration with succinate compared to the other substrates under LG, NG and IHG treatments (S4A Fig). Furthermore, for palmitate we showed a significant difference between LG and HG (P = 0.037 LG vs HG) and for pyruvate we showed a significant difference between LG, HG and IHG (P = 0.017 LG vs HG; P = 0.017 LG vs IHG).

At 72h, we showed no difference between glucose treatments (LG, NG, HG and IHG) on the mitochondrial respiration whatever the substrate used (S4B Fig). However, we observed a higher mitochondrial respiration with succinate compared to the other substrates under LG (P < 0.001 succinate vs palmitate, P = 0.005 succinate vs pyruvate) NG (P < 0.001 succinate vs palmitate, P < 0.001 succinate vs pyruvate) and HG (P < 0.001 succinate vs palmitate, P < 0.001 succinate vs pyruvate) treatments but not IHG.

Comparison between 12 and 72h treatments revealed a higher oxygen consumption for succinate under LG (P < 0.001), NG (P = 0.003) and HG (P = 0.004) but not with IHG treatment and for pyruvate only under LG (P = 0.031) treatment at 72h (Fig 4A–4C).

## Effect of glucose treatment on ATP production

ATP production was measured under basal and stimulated conditions (pyruvate + malate or succinate + rotenone) in HL-1 cells pretreated with the different glucose concentrations (LG, NG, HG or IHG) for either 12 or 72h. While at 12h, we found no effect of either glucose treatment nor substrates on ATP production (S5A Fig), we observed a significantly higher ATP levels for LG condition in the presence of succinate + rotenone compared to pyruvate + malate (P < 0.001) and compared to NG, HG and IHG in the presence of succinate + rotenone (P < 0.001 vs NG, HG and IHG) (S5B Fig).

Comparison between 12 and 72h revealed a significant decrease in ATP production for LG (P = 0.008) and HG (P = 0.028) treatments in the presence of pyruvate + malate at 72h and a

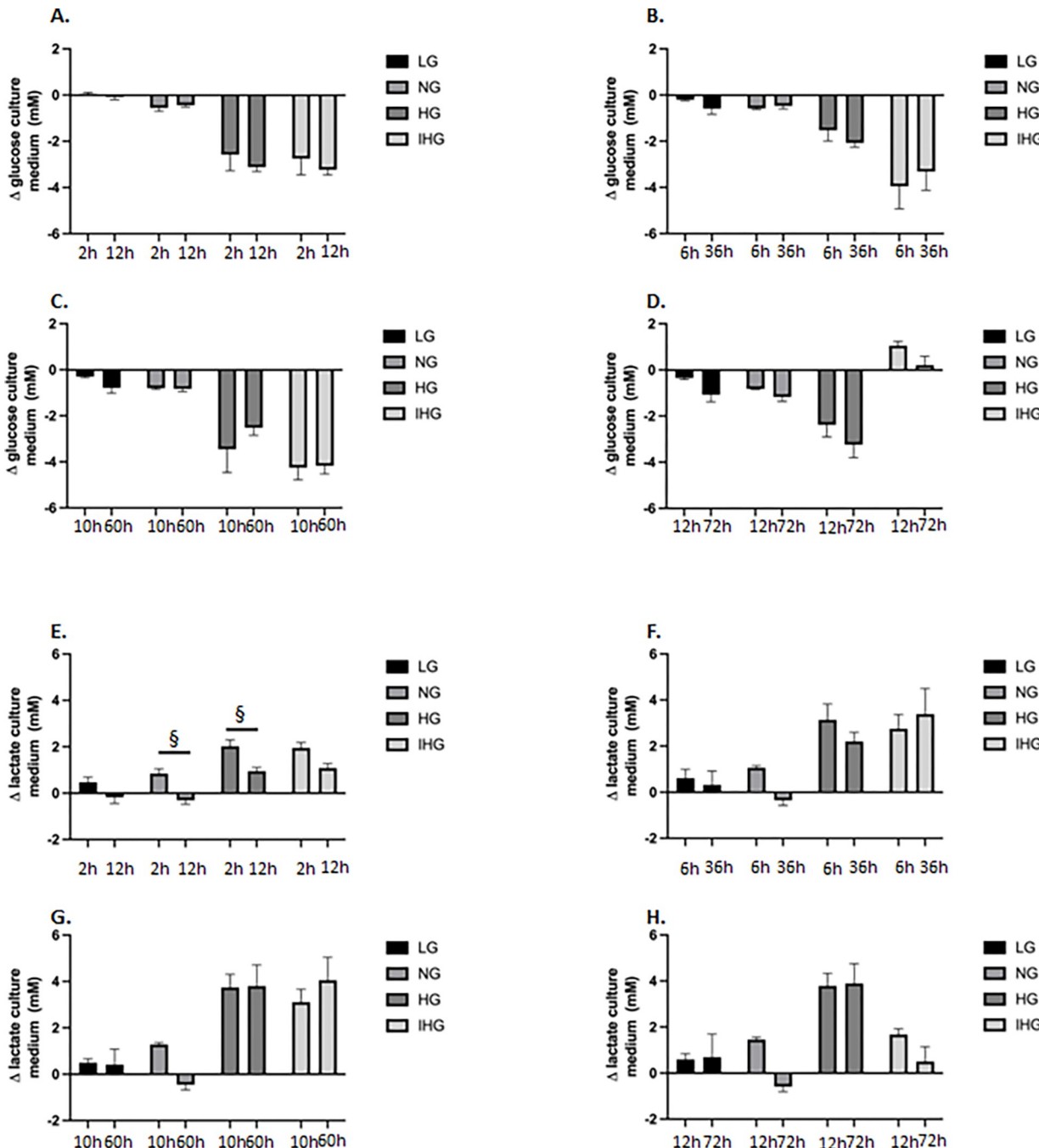

**Fig 3. Comparison of glucose consumption and lactate production in culture medium of HL-1 cells exposed to LG, NG, HG and IHG between short- (12h) and long-time (72h) glucose treatment.** HL-1 cells were cultured at least during 3 weeks with normal glucose (5.5 mmol/l) then submitted to LG, NG, HG or IHG for 12h with medium change every 2h or 72h with medium change every 12h (see Fig 1). Glucose (A, B, C and D) and lactate (E, F, G and H) concentrations were measured in the culture medium at different times (see Fig 1). Data are the means ± S.E.M of 4–6 independent experiments. Two-way ANOVA followed by the Tukey's multiple comparisons test when evaluating the effect of glucose and time treatment. *, P < 0.05, short- vs long-time treatment. For glucose consumption, $F_{2 \text{ vs } 12h}$ (1, 32) = 1.362, P = 0.252; $F_{glucose\ treatment}$ (3, 32) = 48.340, P<0.0001; $F_{2 \text{ vs } 12h \times glucose\ treatment}$ (3, 32) = 0.480, P = 0.698; $F_{6 \text{ vs } 36h}$ (1, 32) = 0.017, P = 0.895; $F_{glucose\ treatment}$ (3, 32) = 18.82, P<0.0001; $F_{6 \text{ vs } 36h \times glucose\ treatment}$ (3, 32) = 0.584, P = 0.629; $F_{10 \text{ vs } 60h}$ (1, 32) = 0.190, P = 0.665; $F_{glucose\ treatment}$ (3, 32) = 37.83, P<0.0001; $F_{10 \text{ vs } 60h \times glucose\ treatment}$ (3, 32) = 1.069, P = 0.376; $F_{12 \text{ vs } 72h}$ (1, 31) = 7.478, P = 0.010; $F_{glucose\ treatment}$ (3, 31) = 30.51, P<0.0001; $F_{12 \text{ vs } 72h \times glucose\ treatment}$ (3, 31) = 0.223, P = 0.8792. For lactate production, $F_{2 \text{ vs } 12h}$ (1, 32) = 34.96, P<0.0001; $F_{glucose\ treatment}$ (3, 32) = 22.41, P<0.0001; $F_{2 \text{ vs } 12h \times glucose\ treatment}$ (3, 32) = 0.500, P = 0.684; $F_{6 \text{ vs } 36h}$ (1, 32) = 1.186, P = 0.284; $F_{glucose\ treatment}$ (3, 32) = 9.598, P = 0.0001; $F_{6 \text{ vs } 36h \times glucose\ treatment}$ (3, 32) = 0.918, P = 0.442; $F_{10 \text{ vs } 60h}$ F (1, 32) = 0.160, P = 0.691; $F_{glucose\ treatment}$ (3, 32) = 13.93, P<0.0001; $F_{10 \text{ vs } 60h \times glucose\ treatment}$ F (3, 32) = 1.247, P = 0.309; $F_{12 \text{ vs } 72h}$ (1, 31) = 2.582, P = 0.118; $F_{glucose\ treatment}$ (3, 31) = 11.01, P<0.0001; $F_{12 \text{ vs } 72h \times glucose\ treatment}$ (3, 31) = 1.254, P = 0.307.

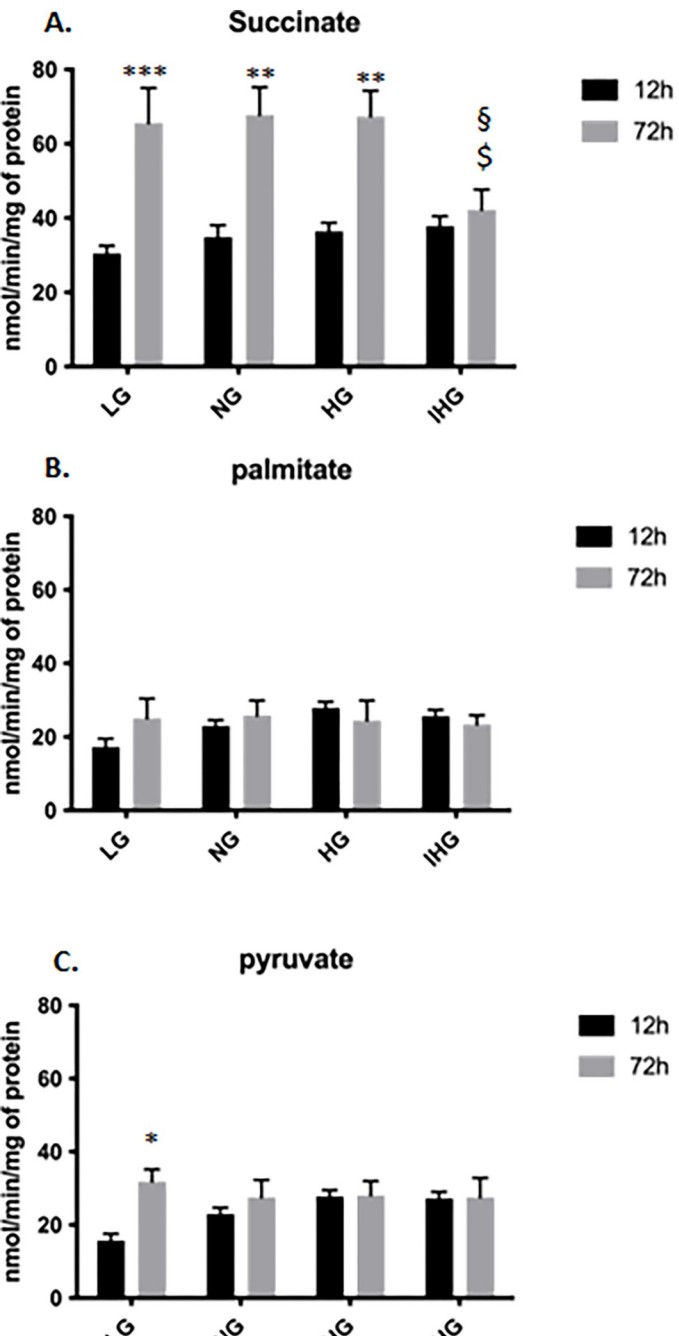

**Fig 4. Comparison of mitochondrial respiration of HL-1 cells exposed to LG, NG, HG and IHG between short- (12h) and long-time (72h) glucose treatment.** HL-1 cells were cultured at least during 3 weeks with normal glucose (5.5 mmol/l) then submitted to LG, NG, HG or IHG for 12h with medium change every 2h or 72h with medium change every 12h (see Fig 1). Oxygen consumption was measured in the presence of either succinate (A), palmitate (B) or pyruvate (C). Data are the means ± S.E.M of 5–7 independent experiments. Two-way ANOVA followed by the Tukey's multiple comparisons test when evaluating the effect of glucose and time treatment. **, $P < 0.01$, ***, $P < 0.001$, 12 vs 72h treatment. $, $P < 0.05$, vs NG, $, $P < 0.05$, vs HG. For succinate, $F_{12 \text{ vs } 72h} (1, 35) = 47.73$, $P < 0.0001$; $F_{\text{glucose treatment}} (3, 35) = 2.160$, $P = 0.1103$; $F_{12 \text{ vs } 72h \text{X glucose treatment}} (3, 35) = 3.768$, $P = 0.0192$; For palmitate, $F_{12 \text{ vs } 72h} (1, 36) = 0.2745$, $P = 0.6035$; $F_{\text{glucose treatment}} (3, 36) = 0.8219$, $P = 0.4904$; $F_{12 \text{ vs } 72h \text{X glucose treatment}}$ $(3, 36) = 1.261$, $P = 0.3021$; For pyruvate, $F_{12 \text{ vs } 72h} (1, 33) = 5.242$, $P = 0.0286$; $F_{\text{glucose treatment}} (3, 33) = 0.6641$, $P = 0.5801$; $F_{12 \text{ vs } 72h \text{X glucose treatment}} (3, 33) = 2.535$, $P = 0.0737$.

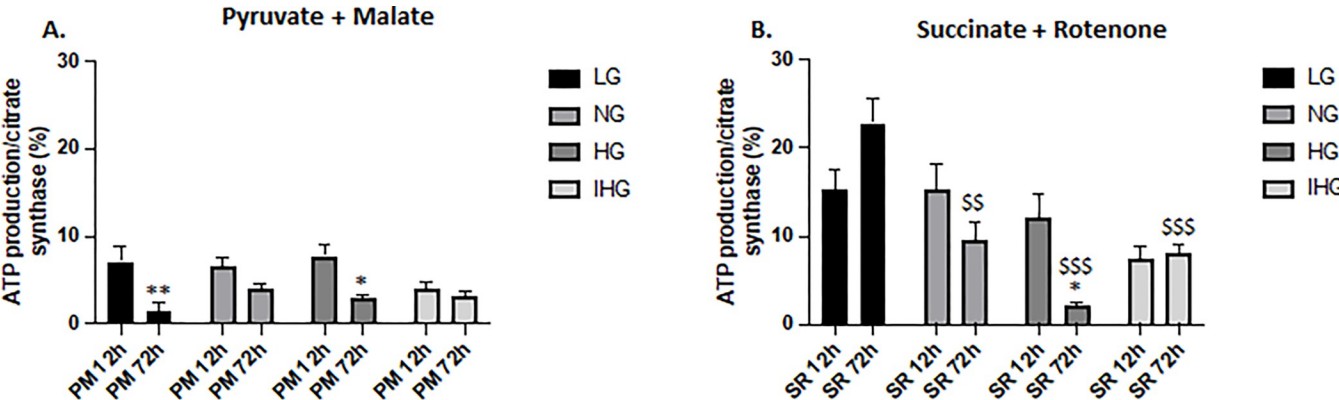

**Fig 5. Comparison of ATP production in HL-1 cells exposed to LG, NG, HG and IHG between short- (12h) and long-time (72h) glucose treatment.** HL-1 cells were cultured at least during 3 weeks with normal glucose (5.5 mmol/l) then submitted to LG, NG, HG or IHG for 12h with medium change every 2h or 72h with medium change every 12h (see Fig 1). ATP production was measured in the absence (basal) or in the presence of either pyruvate + malate (A) or succinate + rotenone (B). Results were normalized to the citrate synthase activity and expressed in %. Data are the means ± S.E.M of 3–5 independent experiments. Two-way ANOVA followed by the Tukey's multiple comparisons test when evaluating the effect of glucose treatment and substrates. *, P < 0.05, **, P < 0.01, 12h vs 72h, $$, P < 0.01, $$$, P < 0.001 vs LG. For pyruvate + malate, $F_{12 \text{ vs } 72h} (1, 24) = 25.920$, P<0.001; $F_{\text{glucose treatment}} (3, 24) = 1.584$, P = 0.219; $F_{12 \text{ vs } 72h \times \text{glucose treatment}} (3, 24) = 2.503$, P = 0.083. For succinate + rotenone, $F_{12 \text{ vs } 72h} (1, 24) = 1.556$, P = 0.224; $F_{\text{glucose treatment}} (3, 24) = 14.51$, P<0.001; $F_{12 \text{ vs } 72h \times \text{glucose treatment}} (3, 24) = 6.737$, P = 0.002.

higher ATP levels for LG compared to the other glucose treatments (P = 0.002 vs NG, P < 0.001 vs HG and IHG) in the presence of succinate + rotenone at 72h (Fig 5A and 5B).

## Effect of glucose treatment on the mitochondrial membrane potential (MMP)

We measured the MMP with the mitochondrial specific dye TMRE under basal and stimulated conditions (succinate, pyruvate or palmitate) in HL-1 cells pretreated with the different glucose concentrations (LG, NG, HG or IHG) for either 12 or 72h. First, we observed that CCCP, a well-described uncoupling agent of the mitochondrial oxidative phosphorylation, significantly decreased MMP by more than 50% (3025 ±85 FU for control vs 1436 ±186 FU for CCCP, P = 0.002). While at 12h, we didn't observe significant difference in the absence (basal) or in the presence of substrates under LG, NG, HG or IHG treatments (S6A Fig), we showed a significant difference between LG, NG and IHG for succinate (P = 0.047 LG vs IHG and P = 0.045 NG vs IHG), and between LG and HG for palmitate (P = 0.042 LG vs HG) (S6B Fig).

Comparison between 12 and 72h revealed a significant decrease of the MMP for basal under HG (P = 0.041) and IHG (P < 0.001) treatments (Fig 6A–6D).

## Effect of glucose treatment on the mitochondrial superoxide anion production and on protein oxidation

In the last part of our study, we examined whether glucose treatments would increase reactive oxygen species (ROS) production and oxidative stress. HL-1 cells were exposed to the different glucose concentrations (LG, NG, HG or IHG) either for 12 or 72h followed by incubation with MitoSOX probe, to measure mitochondrial superoxide anion production in basal conditions and in the presence of antimycin A, a specific complex III inhibitor, which is known to increase ROS level [12]. At 12h, antimycin A increased superoxide anion production by 30 to 50% in the different glucose conditions (mean fold increase for all treatments 1.38+/-0.04) while at 72h, this increase ranged from 57 to 74% (mean fold increase for all treatments 1.65 +/-0.04).

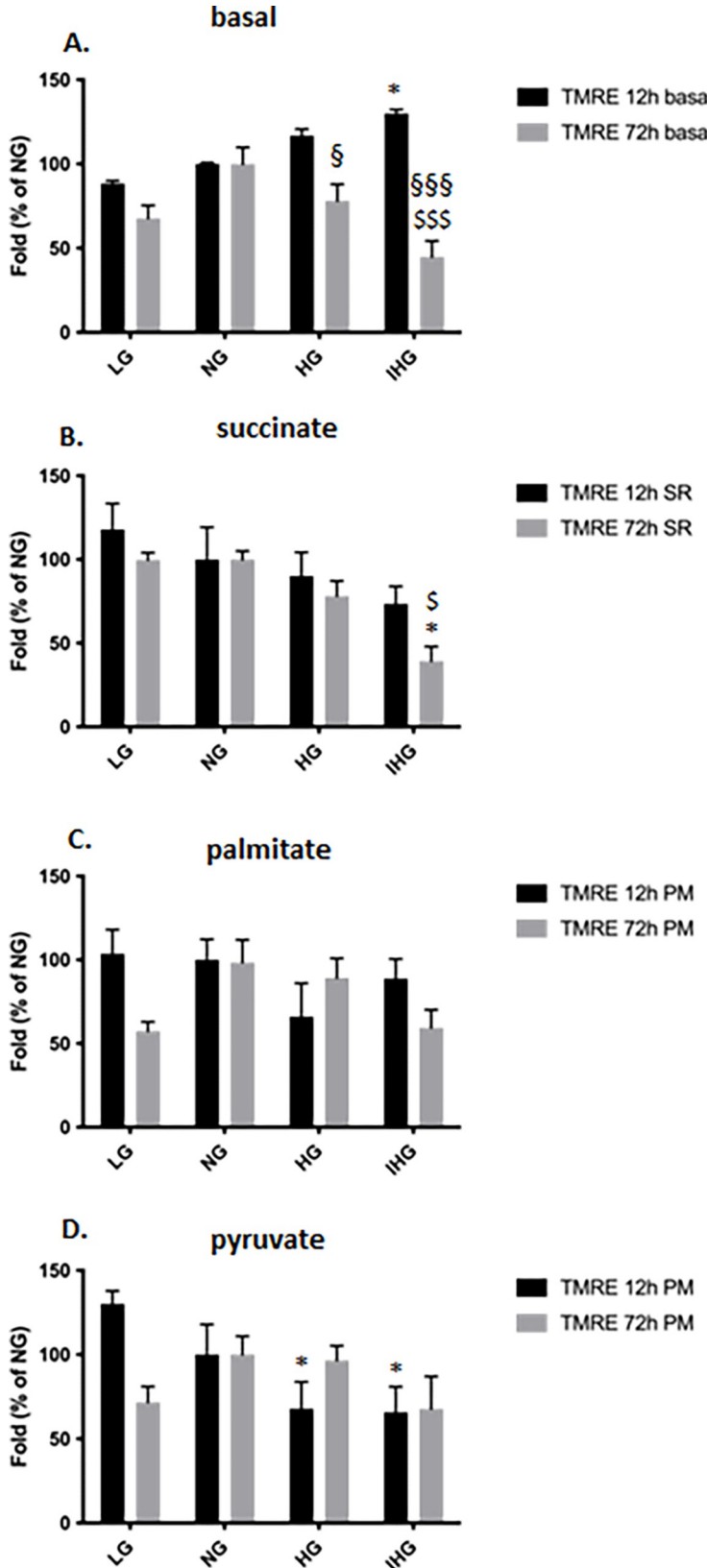

**Fig 6. Comparison of the MMP in HL-1 cells exposed to LG, NG, HG and IHG between short- (12h) and long-time (72h) glucose treatment.** HL-1 cells were cultured at least during 3 weeks with normal glucose (5.5 mmol/l) then submitted to LG, NG, HG or IHG for 12h with medium change every 2h or 72h with medium change every 12h (see Fig 1). The MMP was measured using TMRE in the absence (basal, A) or in the presence of either succinate (B), palmitate (C) or pyruvate (D). Results were normalized to the NG condition. Data are the means ± S.E.M of 3–7 independent experiments. Two-way ANOVA followed by the Tukey's multiple comparisons test when evaluating the effect of glucose and time treatment *, P < 0.05, vs LG; $, P < 0.05, $$$, P < 0.001, vs NG; $, P < 0.05, §§§, P < 0.001, 12 vs 72h. For basal, $F_{12 \text{ vs } 72h}$ (1, 20) = 41.98, P<0.0001; $F_{\text{glucose treatment}}$ (3, 20) = 3.239, P = 0.0438; $F_{12 \text{ vs } 72h \text{X glucose treatment}}$ (3, 20) = 10.56, P = 0.0002; For succinate, $F_{12 \text{ vs } 72h}$ (1, 32) = 3.722, P = 0.0626; $F_{\text{glucose treatment}}$ (3, 32) = 7.572, P = 0.0006; $F_{12 \text{ vs } 72h \text{X glucose treatment}}$ (3, 32) = 0.7248, P = 0.5447; For palmitate, $F_{12 \text{ vs } 72h}$ (1, 36) = 2.257, P = 0.1417; $F_{\text{glucose treatment}}$ (3, 36) = 1.543, P = 0.2200; $F_{12 \text{ vs } 72h \text{X glucose treatment}}$ (3, 36) = 2.886, P = 0.0489; For pyruvate, $F_{12 \text{ vs } 72h}$ (1, 28) = 0.4816, P = 0.4934; $F_{\text{glucose treatment}}$ (3, 28) = 2.624, P = 0.0701; $F_{12 \text{ vs } 72h \text{X glucose treatment}}$ (3, 28) = 3.349, P = 0.0331.

Both at 12 and 72h, we observed that HG and IHG significantly increased superoxide anion production compared to LG (for the basal condition; at 12h, P < 0.001 LG vs HG and P < 0.001 LG vs IHG; at 72h, P < 0.001 LG vs HG and P < 0.001 LG vs IHG) and NG (for the basal condition; at 12h, P < 0.001 NG vs HG and P < 0.001 NG vs IHG; at 72h, P < 0.001 NG vs HG and P < 0.001 NG vs IHG) in the absence and in the presence of antimycin A (for antimycin A condition; at 12h, P < 0.001 LG vs HG, P = 0.008 LG vs IHG, P < 0.001 NG vs HG and P = 0.001 NG vs IHG; at 72h, P < 0.001 LG vs HG and P < 0.001 LG vs IHG, P < 0.001 NG vs HG and P < 0.001 NG vs IHG) (S7A and S7B Fig). Surprisingly, we also showed a modest but higher superoxide anion production upon LG compared to NG but only in the basal condition at 12h (P = 0.029 LG vs NG) (S7A Fig).

When reported to NG condition and as depicted in the Fig 7A and 7B, we observed a significant decrease of the superoxide anion production at 72h under LG treatment in the absence and in the presence of antimycin A (P = 0.001 for basal, P = 0.001 for antimycin A). In contrast, IHG treatment increased superoxide anion production at 72h compared to 12h (P < 0.001) in the presence of the complex III inhibitor (Fig 7B).

We further evaluated whether glucose treatments during 12 or 72h increased protein oxidation by the oxyblot assay. While no significant modification was detected with LG, NG, HG or

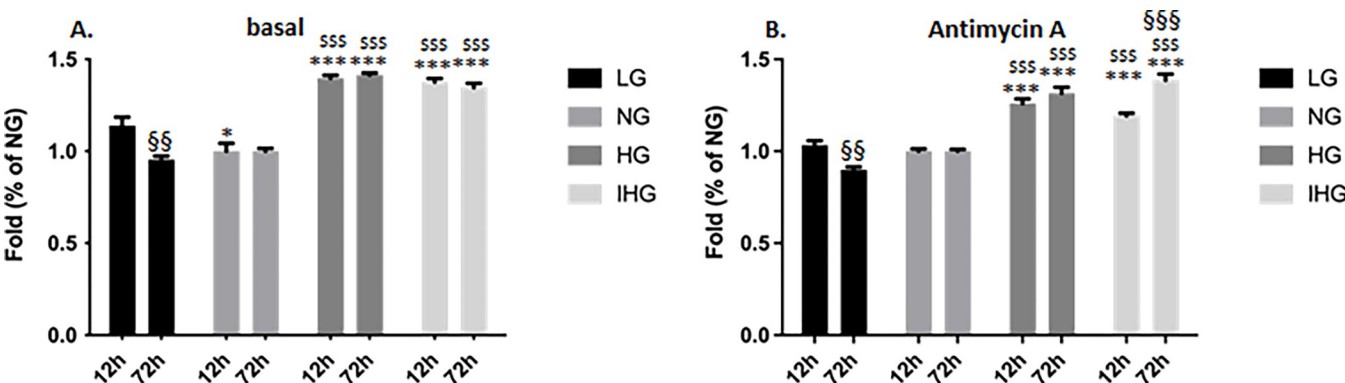

**Fig 7. Measurement of the mitochondrial superoxide anion production in HL-1 cells exposed to LG, NG, HG and IHG between short- (12h) and long-time (72h) glucose treatment.** HL-1 cells were cultured at least during 3 weeks with normal glucose (5.5 mmol/l) then submitted to LG, NG, HG or IHG for 12h with medium change every 2h or 72h with medium change every 12h (see Fig 1). The mitochondrial superoxide anion production was measured using MitoSox under basal (A) or in the presence of the specific complex III inhibitor, antimycin A (B). Results were normalized to the NG condition. Data are the means ± S.E.M of 3–4 independent experiments. Two-way ANOVA followed by the Tukey's multiple comparisons test when evaluating the effect of glucose and time treatment *, P < 0.05, ***, P < 0.001, vs LG; $$$, P < 0.001, vs NG; §§, P < 0.01§§§, P < 0.001, 12 vs 72h. For basal, $F_{12 \text{ vs } 72h}$ (1, 20) = 5.273, P = 0.0326; $F_{\text{glucose treatment}}$ (3, 20) = 96.49, P<0.0001; $F_{12 \text{ vs } 72h \text{X glucose treatment}}$ (3, 20) = 4.705, P = 0.0121; For antimycin A, $F_{12 \text{ vs } 72h}$ (1, 20) = 3.412, P = 0.0796; $F_{\text{glucose treatment}}$ (3, 20) = 127.5, P<0.0001; $F_{12 \text{ vs } 72h \text{X glucose treatment}}$ (3, 20) = 18.76, P<0.0001.

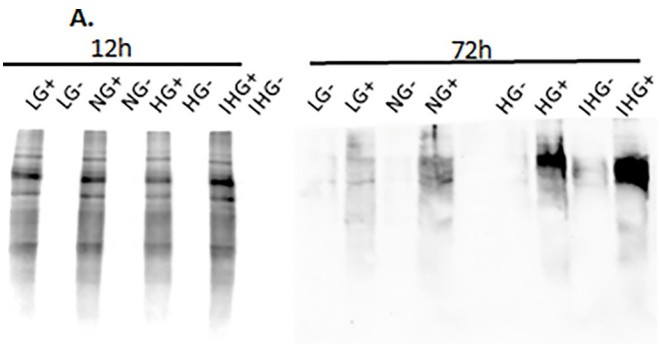
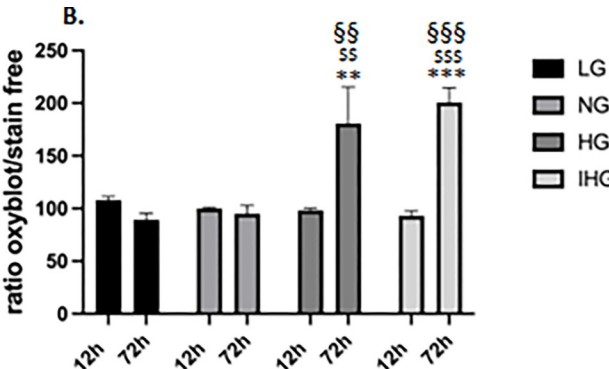

**Fig 8. Comparison of oxidized proteins extracted from HL-1 cells exposed to LG, NG, HG and IHG between short- (12h) and long-time (72h) glucose treatment.** HL-1 cells were cultured at least during 3 weeks with normal glucose (5.5 mmol/l) then submitted to LG, NG, HG or IHG for 12h with medium change every 2h or 72h with medium change every 12h (see Fig 1). Proteins were derivatized (+) or not (-) with DNPH then separated by SDS-PAGE. Total proteins were revealed in free-stain gel and oxidized proteins were revealed by anti-DNP antibody (A). Immunoreactive proteins were quantified by densitometric analysis using Fiji 1.0 and normalized to total stained protein. Data are the means ± S.E.M of 5 independent experiments. Two-way ANOVA followed by the Tukey's multiple comparisons test when evaluating the effect of glucose and time treatment **, $P < 0.01$, ***, $P < 0.001$, vs LG; $^{\$\$}$, $P < 0.01$,$^{\$\$\$}$, $P < 0.001$, vs NG; $^{\S\S}$, $P < 0.01$$^{\S\S\S}$, $P < 0.001$, 12 vs 72h. $F_{12\ vs\ 72h} (1, 32) = 18.15$, $P = 0.0002$; $F_{glucose\ treatment} (3, 32) = 7.220$, $P = 0.0008$; $F_{12\ vs\ 72hXglucose\ treatment} (3, 32) = 10.39$, $P < 0.0001$.

IHG at 12h, we observed a significant increase in protein oxidation for HG and IHG at 72h (P = 0,002) (S8A and S8B Fig).

As depicted in the Fig 8A and 8B, comparison between 12 and 72h revealed significant higher oxidized proteins for HG and IHG treatments (P = 0.004 for HG and P < 0.001for IHG).

## Discussion

### Validation model

The main goal of our study was to evaluate whether IHG, which mimics glucose excursion observed in poorly controlled diabetes, was deleterious for mitochondria functions in the cardiac muscle cell line HL-1. First, we showed that HL-1 cells responded to insulin exposure as evidenced by a decrease of glucose concentration in the culture medium and an increase in lactate production. Our data are in good agreement with the expression of glucose transporter GLUT-1 and 4 in this cell line and the ability of insulin to stimulate glucose uptake [13,14]. The weak increase of glucose consumption mediated by the addition of insulin is probably due to the presence of this hormone in the Claycomb's medium [15].

Effect of IHG exposure.

When HL-1 cells were exposed to IHG, we observed a metabolic impact with higher glucose consumption and lactate production compared to LG and NG both at 12 and 72h treatment. Such metabolic effects were also reported in rat primary cardiomyocytes [9], in the cardiomyocyte cell line H9C2 [16] and in human myotubes exposed to high glucose during 2 to 4 days [17]. Lactate production would result both from glycolysis stimulation and a reduction of lactate oxidation [17].

In polarographic experiments, we showed that IHG treatment increased mitochondrial respiration only after 12h treatment upon pyruvate stimulation but without any effect after 72h treatment. This would suggest a regulation only upon short-time treatment either on the mitochondrial pyruvate carrier or the pyruvate dehydrogenase complex by IHG since this treatment had no effect when using other substrate (i.e. succinate or palmitate). However, we observed that all glucose treatments during 72h increased mitochondrial respiration upon succinate except IHG exposure. Furthermore, when using other substrates (i.e. palmitate and

pyruvate), we did not observe any significant change between 12 and 72h treatments. These data suggest a selective regulation at the complex II by IHG. Succinate dehydrogenase activity is regulated by post-translationnal modifications (phosphorylation, acetylation) [18] and by oxaloacetate [19]. Our data are in good agreement with the report of Bhansali et al. who demonstrated a reduced activity of complex II in peripheral blood mononuclear cell of patients with type 2 diabetes mellitus [20].

When measuring ATP production, we only noted a decrease from IHG treated cells compared to LG exposure at 72h upon succinate stimulation. Such difference could be explained by an increase in mitochondrial respiration upon LG with succinate. However, high $O_2$ consumption was also measured after NG and HG treatments without significant difference with IHG.

After 72h, IHG was shown to decrease the MMP upon succinate stimulation compared to both LG and NG. High glucose exposure was also reported to promote a reduction of MMP in H9c2 cells (50 mmol/l glucose during 24h) [21] and in isolated neonatal rat cardiomyocytes (15 mmol/l glucose during 24h) [22]. The decrease of MMP observed upon ~~HG and~~ IHG could be either due to a reduction of the electron flux in the respiratory chain or an uncoupling of OXPHOS but our data are not consistent with this hypothesis since both conditions would promote either a decrease or an increase of the mitochondrial respiration when using the different substrates (succinate, palmitate and pyruvate), respectively; however, we only observed a low mitochondrial respiration upon succinate but not with palmitate or pyruvate in IHG compared to LG, NG and HG. We can hypothesize that the mitochondrial membrane depolarization could be the consequence of a transient opening of the mitochondrial permeability transition pore (mPTP) regulating the concentration of the mitochondrial calcium homeostasis and the inner mitochondrial membrane voltage [23]. Using the rat kidney proximal tubular cells NRK-52E, Liu and collaborators showed that 33 mmol/l glucose during 48h promotes mPTP opening and a depolarization of the MMP [24]. The lower ATP level observed in IHG compared to LG upon succinate stimulation could be linked to the decrease of the MMP that would reflect an uncoupling of the respiratory chain.

We observed that both IHG and HG after 12 and 72h treatment were able to stimulate mitochondrial ROS production but increase in protein oxidation was only detected after 72h exposure. As we observed higher superoxide anion production after 72h than 12h exposure, we can suppose that a sufficient amount of ROS is required to detect protein oxidation. In the kidney epithelial cells MDCK (Madin-Darby Canine Kidney), both increase in ROS production and in oxidized mitochondrial proteins were detected after 96h exposure to 25 mmol/l glucose [25]. High glucose exposure was also demonstrated to increase ROS production in different cellular models including neonatal rat cardiomyocytes [22] and H9c2 cells [24].

Furthermore, IHG (swing between 5.5 to 30 mmol/l every 2h during 58h) was reported to promote ROS production to a significant greater level than HG [8] while in our cellular model and in our experimental conditions, no difference between HG and IHG was detected. Under high glucose exposure, ROS generation would result from increasing the reducing equivalents for the mitochondrial respiratory chain and consequently promoting mitochondrial membrane hyperpolarization and anion superoxide production [26]; however, such mechanism is unlikely to occur in our conditions since we rather observed a decrease in the MMP. Mitochondrial ROS production could also result from the reverse electron transport (see for review [27]).

## Effect of time exposure

Since the time exposure of cells to glucose treatment in vitro is rather empirical, we have chosen to increase time from 2 to 12h between high and low glucose swing resulting in short-

(12h) or long-time exposure (72h) (Fig 1). We showed that lactate production was significantly higher at 2h compared to 12h for NG and HG. This suggests that long-time exposure would enable a better metabolic adaptation with either a reduction of glucose consumption by the anaerobic glycolysis or a shift from the anaerobic glycolysis to a complete glucose degradation by the Krebs cycle and the mitochondrial respiratory chain. The first hypothesis is unlikely since we did not observe difference of glucose consumption between 2 and 12h treatment. Except for IHG, we observed an increase in oxygen consumption only upon succinate and not with pyruvate and palmitate at 72h compared to 12h suggesting that LG, NG and HG would increase complex II activity. However, this increase in mitochondrial respiration upon succinate was not mirrored by a higher ATP level at 72h compared to short time treatment. Alcantar-Fernandez and collaborators reported that high glucose exposition to C. elegans promoted increased by 3- to 5-fold activity of succinate dehydrogenase [28] which could involve PTPMT1, a mitochondrial tyrosine phosphatase, as a major regulator of complex II activity [29]. Comparison between short and long-time treatment revealed a significant decrease for the MMP upon HG and IHG and an increase both for ROS production and protein oxidation upon IHG. At least concerning IHG, it is tempting to speculate that reduction of electron flux from complex II would be responsible for the decrease in MMP, which would promote ROS production and increase oxidative stress as reviewed [30].

## Limit of the study

The goal of our study was to examine the impact of glucose fluctuation on mitochondria in cardiomyocytes since diabetes mellitus is a well-known cause of cardiomyopathy whose complex pathophysiology would involve mitochondria [31]. The mitochondrial involvement in diabetic cardiomyopathy is currently highlighted since the cardiac benefits of Sodium Glucose Cotransporter 2 inhibitors (SGLT2-i) could be associated with a modulation of mitochondrial function [32]. The strength of our study is the precise control of the glucose concentration and fluctuation compared to experimental animal models of diabetes where classification in the hyperglycemia or in the glucose fluctuation groups remains challenging. Furthermore, this in-vitro study allowed us to assess many cellular parameters and mitochondrial functions that are more difficult to explore in animal models. However, there are also several drawbacks: we arbitrarily chose experimental conditions (glucose concentrations, time of exposure) to mimic what happens in patients with diabetes but extrapolations to patients would be hazardous. For practical reasons, we selected two duration treatments (i.e. 12 and 72h) but it clearly does not reflect clinical situations where glucose fluctuations last over longer time. While we evidenced that IHG treatment promoted mitochondrial dysfunctions, we also observed that HG exposure induced changes indicating that both treatments are deleterious.

Concerning glucose variability and as mentioned by others [33], time exposure is a crucial parameter that clearly influences deleterious effects of IHG treatment on mitochondria, since extension of treatment from 12 to 72h increased mitochondrial dysfunctions. While fatty acids are the main substrates for cardiomyocytes, our culture conditions are not reflecting what happens in type 2 diabetes since fatty acids are only provided by diluted serum. Moreover, in vitro studies do not allow to study the endocrine regulation during the transition from the fed to the fasting state and notably the metabolic switch between the different energy substrates.

In summary, we demonstrated that exposition of HL-1 cells to IHG promoted alterations in mitochondrial respiration and membrane potential, increased ROS production and oxidative stress. Glucose fluctuation could be involved in cardiac complications of diabetes through mitochondrial dysfunctions and as hyperglycemia, glycemic variability should be considered in the management of diabetes.

To go further in the molecular mechanisms involved in IHG-induced mitochondrial dysfunction, we could measure the mitochondrial permeability transition pore and mitochondrial $Ca^{2+}$ during exposure to different concentrations of glucose, which are major regulators of the electrons flux through the respiratory chain. We could also conduct transcriptomic experiments to examine the regulation on genes encoding for enzymes implicated in the metabolism (beta-oxidation, Krebs, respiratory chain complexes. . .) and in ROS detoxification.

## Supporting information

**S1 Fig. Evaluation of glucose consumption and lactate production in culture medium of HL-1 cells exposed to LG, NG, HG and IHG during 12 and 72h.** HL-1 cells were cultured at least during 3 weeks with normal glucose (5.5 mmol/l) then submitted to LG, NG, HG or IHG for 12 or 72h. Glucose (A and C) and lactate (B and D) concentrations were measured in the culture medium at different times (see Fig 1) to calculate glucose consumption and lactate production. Data are the means ± S.E.M of 4–6 independent experiments. Two-way ANOVA followed by the Tukey's multiple comparisons test when evaluating the effect of glucose and time treatment. *, P < 0.05, **, P < 0.01, ***, P < 0.001, vs LG; [$], P < 0.05, [$$], P < 0.01, [$$$], P < 0.001 vs NG; [$], P < 0.05, [$$$], P < 0.001 vs HG. For 12h treatment and glucose consumption, $F_{treatment}$ (3, 48) = 23.85, P<0.0001; $F_{time}$ (3, 48) = 7.03, P = 0.0005; $F_{treatmentXtime}$ (9, 48) = 6.92, P<0.0001. For 12h treatment and lactate production, $F_{treatment}$ (3, 48) = 38.90, P<0.0001; $F_{time}$ (3, 48) = 3.43, P = 0.0242; $F_{treatmentXtime}$ (9, 48) = 1.63, P = 0.132. For 72h treatment and glucose consumption, $F_{treatment}$ (3, 79) = 49.66, P<0.001; $F_{time}$ (3, 79) = 3.602, P = 0.017; $F_{treatmentXtime}$ (9, 79) = 12.26, P<0.001. For 72h treatment and lactate production, $F_{treatment}$ (3, 79) = 22.69, P<0.001; $F_{time}$ (3, 79) = 4.308, P = 0.007; $F_{treatmentXtime}$ (9, 79) = 2.819, P = 0.006. (PPTX)

**S2 Fig. Evaluation of medium glucose consumption in HL-1 cells cultured with high and normal glucose.** HL-1 cells were cultured at least during 3 weeks either with normal (5.5 mmol/l) or high (25 mmol/l) glucose then submitted to 4 different regimens for 12h: LG, NG, HG or IHG. Glucose concentration was measured in the culture medium at 0, 2, 6, 10 and 12h to calculate glucose consumption. Data are the means ± S.E.M of 4 independent experiments. Two-way ANOVA followed by the Tukey's multiple comparisons test when evaluating the effect of time treatment in HL-1 cells culture in normal or high glucose. *, P < 0.05, **, P < 0.01, vs 2h treatment, [$], P < 0.05, vs 12h treatment. For LG treatment, $F_{treatment}$ (3, 24) = 10.31, P = 0.0002; $F_{time}$ (1, 24) = 0.4323, P = 0.5171; $F_{treatmentXtime}$ (3, 24) = 2.24, P = 0.1093. For NG treatment, $F_{treatment}$ (1, 24) = 2.54, P = 0.1241; $F_{time}$ (3, 24) = 6.31, P = 0.0026; $F_{treatmentXtime}$ (3, 24) = 2.85, P = 0.0586. For HG treatment, $F_{treatment}$ (1, 24) = 11.65, P = 0.0023; $F_{time}$ (3, 24) = 0.974, P = 0.4212; $F_{treatmentXtime}$ (3, 24) = 1.12, P = 0.3578. For IHG treatment, $F_{treatment}$ (1, 24) = 7.31, P = 0.0124; $F_{time}$ (3, 24) = 23.63, P<0.0001; $F_{treatmentXtime}$ (3, 24) = 1.43, P = 0.2570. (PPTX)

**S3 Fig. Evaluation of medium lactate production in HL-1 cells culture with high and normal glucose.** HL-1 cells were cultured at least during 3 weeks either with normal (5.5 mmol/l) or high (25 mmol/l) glucose then submitted to 4 different regimens for 12h: LG, NG, HG or IHG. Lactate concentration was measured in the culture medium at 0, 2, 6, 10 and 12h to calculate lactate production. Data are the means ± S.E.M of 4 independent experiments. Two-way ANOVA followed by the Tukey's multiple comparisons test when evaluating the effect of time treatment in HL-1 cells culture in normal or high glucose. For LG treatment, $F_{treatment}$ (1, 24) = 1.09, P = 0.3064; $F_{time}$ (3, 24) = 0.18, P = 0.9057; $F_{treatmentXtime}$ (3, 24) = 0.43, P = 0.7270.

For NG treatment, $F_{treatment}$ (1, 24) = 1.17, P = 0.2883; $F_{time}$ (3, 24) = 1.09, P = 0.3688; $F_{treatmentXtime}$ (3, 24) = 0.03, P = 0.9914. For HG treatment, $F_{treatment}$ "F (1, 24) = 1.89, P = 0.1814; $F_{time}$ (3, 24) = 2.13, P = 0.1222; $F_{treatmentXtime}$ (3, 24) = 0.09, P = 0.9605. For IHG treatment, $F_{treatment}$ (1, 24) = 0.05, P = 0.8237; $F_{time}$ (3, 24) = 1.47, P = 0.2467; $F_{treatmentXtime}$ (3, 24) = 0.20, P = 0.8929.
(PPTX)

**S4 Fig. Mitochondrial respiration of HL-1 cells exposed for 12 or 72h to LG, NG, HG or IHG.** HL-1 cells were cultured at least during 3 weeks with normal (5.5 mmol/l) glucose then submitted to 4 different regimens (LG, NG, HG or IHG) either for 12 (A) or 72h (B). Oxygen consumption was measured by polarography using 3 different substrates. Data are the means ± S.E.M of 5–7 independent experiments. Two-way ANOVA followed by the Tukey's multiple comparisons test when evaluating the effect of substrates or glucose treatments. *, P < 0.05, **, P < 0.01, ***, P < 0.001 vs succinate, $, P < 0.05, vs LG treatment. For 12h treatment, $F_{substrate}$ (2, 56) = 39.66, P<0.0001; $F_{treatment}$ (3, 56) = 13.15, P<0.0001; $F_{substrateXtreatment}$ (6, 56) = 0.453, P = 0.8394. For 72h treatment, $F_{substrate}$ (2, 48) = 48.59, P<0.0001; $F_{treatment}$ (3, 48) = 2.038, P = 0.1210; $F_{substrateXtreatment}$ (6, 48) = 1.443, P = 0.2182.
(PPTX)

**S5 Fig. Measurement of ATP production in HL-1 cells exposed for 12 or 72h to LG, NG, HG or IHG.** HL-1 cells were cultured at least during 3 weeks with normal (5.5 mmol/l) glucose then submitted to 4 different regimens (LG, NG, HG or IHG) either for 12 (A) or 72h (B). The mitochondrial ATP production was measured under basal (no substrate) or stimulated conditions (pyruvate + malate or succinate + rotenone). After subtracting ATP level obtained without stimulation, results were normalized to the activity of the citrate synthase activity and expressed in %. Data are the means ± S.E.M of 3–5 independent experiments. Two-way ANOVA followed by the Tukey's multiple comparisons test when evaluating the effect of substrates or glucose treatments. ***, P < 0.001 vs LG treatment, §§§, P < 0.001 pyruvate + malate vs succinate + rotenone. For 12h treatment, $F_{substrate}$ (1, 22) = 21.000, P = 0.001; $F_{treatment}$ (3, 22) = 3.776, P = 0.025; $F_{substrateXtreatment}$ (3, 22) = 0.984, P = 0.418. For 72h treatment, $F_{substrate}$ (1, 26) = 57.910, P<0.001; $F_{treatment}$ (3, 26) = 15.880, P<0.0001; $F_{substrateXtreatment}$ (3, 26) = 22.200, P<0.001.
(PPTX)

**S6 Fig. Measurement of the MMP in HL-1 cells exposed for 12 or 72h to LG, NG, HG or IHG.** HL-1 cells were cultured at least during 3 weeks with normal (5.5 mmol/l) glucose then submitted to 4 different regimens (LG, NG, HG or IHG) either for 12 (A) or 72h (B). The MMP was measured using TMRE under basal or stimulated conditions (succinate, palmitate or pyruvate). Results were normalized to the NG condition. Data are the means ± S.E.M of 3–7 independent experiments. Two-way ANOVA followed by the Tukey's multiple comparisons test when evaluating the effect of substrates or glucose treatments. *, P < 0.05 vs LG treatment, $, P < 0.05, vs NG treatment. For 12h treatment, $F_{substrate}$ (3, 56) = 1.107, P = 0.3541; $F_{treatment}$ (3, 56) = 2.212, P = 0.0967; $F_{substrateXtreatment}$ (9, 56) = 1.799, P = 0.0887. For 72h treatment, $F_{substrate}$ (3, 64) = 0.5330, P = 0.6613; $F_{treatment}$ (3, 64) = 11.53, P<0.0001; $F_{substrateXtreatment}$ (9, 64) = 1.619, P = 0.1286.
(PPTX)

**S7 Fig. Measurement of mitochondrial superoxide anion production in HL-1 cells exposed for 12 or 72h to LG, NG, HG or IHG.** HL-1 cells were cultured at least during 3 weeks with normal (5.5 mmol/l) glucose then submitted to 4 different regimens (LG, NG, HG or IHG) either for 12 (A) or 72h (B). The mitochondrial superoxide anion production was measured

using MitoSox under basal or in the presence of the specific complex III inhibitor, antimycin A. Results were normalized to the NG condition. Data are the means ± S.E.M of 3–4 independent experiments. Two-way ANOVA followed by the Tukey's multiple comparisons test when evaluating the effect of glucose treatments and antimycin A. *, $P < 0.05$, **, $P < 0.01$, ***, $P < 0.001$ vs LG treatment, $^{\$\$}$, $P < 0.01$, $^{\$\$\$}$, $P < 0.001$ vs NG treatment; $^{\$}$, $P < 0.05$, $^{\$\$}$, $P < 0.01$, basal vs antimycin. For 12h treatment, $F_{antimycin\ A}$ (1, 24) = 29.42, P<0.0001; $F_{treatment}$ (3, 24) = 63.82, P<0.0001; $F_{antimycin\ AXtreatment}$ (3, 24) = 3.914, P = 0.0208. For 72h treatment, $F_{antimycin\ A}$ (1, 16) = 3.569, P = 0.0771; $F_{treatment}$ (3, 16) = 239.6, P<0.0001; $F_{antimycin\ AXtreatment}$ (3, 16) = 4.047, P = 0.0256.
(PPTX)

**S8 Fig. Detection of oxidized proteins by oxyblot in HL-1 cells exposed for 12 or 72h to LG, NG, HG or IHG.** HL-1 cells were cultured at least during 3 weeks with normal (5.5 mmol/l) glucose then submitted to 4 different regimens (LG, NG, HG or IHG) for either 12 (A) or 72h (B). The oxidized proteins were detected with the oxyblot assay. Total proteins were detected in stain free gels and oxidized proteins after derivatization by DNPH. Results were expressed as the ratio of oxidized proteins/total proteins. Data are the means ± S.E.M of 5 independent experiments. One-way ANOVA followed by Dunnett's multiple comparisons test. *, $P < 0.05$, **, $P < 0.01$, vs LG treatment, $^{\$}$, $P < 0.05$, vs NG treatment.
(PPTX)

**S1 Raw data.**
(PDF)

## Acknowledgments

We acknowledge Mr Charlie Lepetit for his technical help regarding TRME assays.

## Author Contributions

**Conceptualization:** Michael Joubert, Stéphane Allouche.

**Formal analysis:** Patrick Mordel, Stéphane Allouche.

**Investigation:** Patrick Mordel, Fanny Fontaine, Quentin Dupas, Stéphane Allouche.

**Methodology:** Patrick Mordel, Fanny Fontaine, Quentin Dupas, Stéphane Allouche.

**Resources:** Stéphane Allouche.

**Visualization:** Patrick Mordel, Stéphane Allouche.

**Writing – original draft:** Patrick Mordel.

**Writing – review & editing:** Michael Joubert, Stéphane Allouche.

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
