## [Decision Letter · Decision Letter 0]

14 Oct 2022

PONE-D-22-24271Glucose fluctuation promotes mitochondrial dysfunctions in the cardiomyocyte cell line HL-1PLOS ONE

Dear Dr. Allouche,

Thank you for submitting your manuscript to PLOS ONE. After careful consideration, we feel that it has merit but does not fully meet PLOS ONE’s publication criteria as it currently stands. Therefore, we invite you to submit a revised version of the manuscript that addresses the points raised during the review process.

We look forward to receiving your revised manuscript.

Kind regards,

Rodrigo Franco

Academic Editor

PLOS ONE

Journal Requirements:

"This work was supported by Université Caen Normandie (Unicaen – France)."

"This work was supported by Université Caen Normandie (Unicaen – France)."

"PM, QD and SA declare that they have no conflict of interest. MJ has received research grants and/or advisory fees from Astrazeneca, Bayer, Boehringer Ingelheim, Lilly and Novonordisk. "

7. We note that you have included the phrase “data not shown” in your manuscript. Unfortunately, this does not meet our data sharing requirements. PLOS does not permit references to inaccessible data. We require that authors provide all relevant data within the paper, Supporting Information files, or in an acceptable, public repository. Please add a citation to support this phrase or upload the data that corresponds with these findings to a stable repository (such as Figshare or Dryad) and provide and URLs, DOIs, or accession numbers that may be used to access these data. Or, if the data are not a core part of the research being presented in your study, we ask that you remove the phrase that refers to these data.

Reviewers' comments:

Reviewer's Responses to Questions

**Comments to the Author**

1. Is the manuscript technically sound, and do the data support the conclusions?

Reviewer #1: Partly

Reviewer #2: No

2. Has the statistical analysis been performed appropriately and rigorously? 

Reviewer #1: Yes

Reviewer #2: No

3. Have the authors made all data underlying the findings in their manuscript fully available?

Reviewer #1: Yes

Reviewer #2: Yes

4. Is the manuscript presented in an intelligible fashion and written in standard English?

Reviewer #1: Yes

Reviewer #2: No

5. Review Comments to the Author

Reviewer #1: The manuscript of Hernandez et al. aims to evaluate the impact of glucose swings on mitochondrial functions in the cardiomyocyte cell line HL-1. Exposure of HL-1 cells to glucose swings promoted time-dependent mitochondrial dysfunctions suggesting a deleterious effect of such condition in patients with diabetes that could contribute to diabetic cardiomyopathy. The manuscript is well written. However, a few comments and questions are provided for clarification and some additional evidence is encouraged.

1. The authors used four different groups such as (LG, 2.8 mmol/l), normal (NG, 5.5 mmol/l), high (HG, 25 mmol/l) or intermittent high glucose (IHG, swing between low and high). It would be good to use another group as no glucose to measure the glucose

consumption.

2. Glucose fluctuation promotes mitochondrial dysfunctions, to confirm that authors need to measure the product ATP levels.

3. There is no molecular mechanism established in this manuscript. The authors need to discuss the future studies warranted to identify the mechanism involving mitochondrial dysfunction promoted by glucose fluctuation.

Reviewer #2: This manuscript discusses how glycemic variability has been suggested as a risk factor for diabetes complications; however, these mechanisms remain poorly understood. The authors demonstrated that glucose swings in HL-1 cells promote time-dependent mitochondrial dysfunctions suggesting a harmful effect of such condition in patients with diabetes that could contribute to diabetic cardiomyopathy.

In figure 2A, the authors do not explain why the graph has negative values and the standard error is overlapping in the insulin treatment and no insulin treatment; it looks like there is no significant difference.

The following graphs are similar, but the authors do not explain why the graphs of glucose concentration have negative values; neither is it clear whether all experiments with glucose were performed in the presence of insulin because most of the patients are treated with insulin. So the cellular model does not fit to compare partially at the human level. The authors did not measure any GLUT transporter, whether oxidized or reduced; GLUT transporter status is not shown, and neither is cell death, which means there was no cell death in the hypoglycemic phase. How was mitochondria function affected to produce superoxide anion instead of water?

The authors should perform WB for Glut-1 and 4 to see which one is involved and fluorescent microscopy to see how cells are affected morphologically.

At this point, the manuscript is very confusing and hard to follow.

6. PLOS authors have the option to publish the peer review history of their article (what does this mean?). If published, this will include your full peer review and any attached files.

Reviewer #1: **Yes: **Anandhan Annadurai

Reviewer #2: No

---

## [Author Response · Author response to Decision Letter 0]

1 Mar 2023

Response to Reviewers

Caen, France 23/02/2023

 Dear Editor, 

Thank you for giving us the opportunity to submit a revised version of our manuscript entitled “Glucose fluctuation promotes mitochondrial dysfunctions in the cardiomyocyte cell line HL-1” by Mordel and collaborators. We also would like to thank reviewers for their useful and detailed comments that have improved the quality of our manuscript.

Responses to the Editor :

Point 1 : We have made changes as requested in the revised version.

Points 2 and 3: Concerning the Financial Disclosure, we had no grant numbers for the awards we received from the Université Caen Normandie (Unicaen – France). We don’t know where to mention this information.

Point 4 : We have updated the statement about conflict of interest in our revised manuscript as follows :

Conflict of interest: PM, FF, QD and SA declare that they have no conflict of interest. MJ has received research grants and/or advisory fees from Astrazeneca, Bayer, Boehringer Ingelheim, Lilly and Novonordisk. This does not alter our adherence to PLOS ONE policies on sharing data and materials.

Point 5 : Upon re-submitting our revised manuscript, we upload all data obtained and presented in this study.

Point 6 : Western blot and gel image data were included in the supporting information file.

Point 7 : We removed the phrase “data not shown” in the revised manuscript and replaced by data. All the data obtained in the present work were included in the supporting information file.

Responses to the Reviewer #1: 

1. The authors used four different groups such as (LG, 2.8 mmol/l), normal (NG, 5.5 mmol/l), high (HG, 25 mmol/l) or intermittent high glucose (IHG, swing between low and high). It would be good to use another group as no glucose to measure the glucose consumption.

Response : A total glucose withdrawal from the culture medium would have been too deleterious for HL-1 cells to conduct experiments mainly during 72h. Most of the cell lines used in labs, including HL-1 cells, are highly dependent on glucose oxidation and culture in glucose-free medium leads to a major reduction in cell proliferation and in cell death. This is the main reason why we did not include this group “no glucose” in our experiments. In addition, we wanted to mimic physiological or pathological settings with our experimental conditions, what a free-glucose group would have not allowed.

2. Glucose fluctuation promotes mitochondrial dysfunctions, to confirm that authors need to measure the product ATP levels.

As recommended by the reviewer, we conducted experiments to measure ATP production upon 12 and 72h treatment under the different conditions : LG, NG, HG and IHG. Those data are included in the revised version.

3. There is no molecular mechanism established in this manuscript. The authors need to discuss the future studies warranted to identify the mechanism involving mitochondrial dysfunction promoted by glucose fluctuation.

Response : We agree with the reviewer’s comment. Our data are mainly descriptive and the molecular mechanisms involved in mitochondrial dysfunction upon IHG are speculative. 

We think that complementary experiments would be necessary to go further in those mechanisms. For instance, we could measure the mitochondrial permeability transition pore and mitochondrial Ca2+ during exposure to different concentrations of glucose, which are major regulators of the electrons flux through the respiratory chain. We could also conduct transcriptomic experiments to examine the regulation of genes encoding for enzymes implicated in the metabolism (beta-oxidation, Krebs, respiratory chain complexes…) and in ROS detoxification. 

As suggested by the reviewer, we added this short paragraph in the perspective of this work.

Responses to the Reviewer #2: 

1. In figure 2A, the authors do not explain why the graph has negative values and the standard error is overlapping in the insulin treatment and no insulin treatment; it looks like there is no significant difference.

Response : Values in the fig 2A are negative since we observed glucose consumption by HL-1 cells. As indicated in the materials and methods, we measured glucose concentrations in culture medium at t=0 and t=12 or 72h then we determined glucose consumption as follows : glucose t=12 or 72h - glucose t=0.

We agree with the reviewer that values obtained with and without insulin are overlapping but statistical analysis using the Wilcoxon test indicated a significant difference between the 2 groups (-4.84 +/- 0.75 vs -6.01+/-0.82; p = 0.031). In previous papers, insulin was demonstrated to promote glucose uptake but with more sensitive methods (3H-deoxyglucose) (Chaudary et al., 2002; Shuralyova et al., 2004).

2. The following graphs are similar, but the authors do not explain why the graphs of glucose concentration have negative values; neither is it clear whether all experiments with glucose were performed in the presence of insulin because most of the patients are treated with insulin. 

Response : Except for experiments showed in the Fig.2A and B, we did not add additional insulin in the culture medium. As described by White et al. (2004), the Claycomb medium has already human recombinant insulin (15 𝜇g/l) and this could explain why we did not obtain a huge glucose consumption when we added further insulin. This point was added to the discussion.

3. So the cellular model does not fit to compare partially at the human level. 

Response : When using an experimental model, we should keep in mind that it has limitations to represent a pathophysiological state in humans. However, to elucidate molecular mechanisms they are very useful since we can precisely place cells in specific conditions (ex : a precise glucose swing between high and low glucose concentration). This point was clearly indicated in the discussion (lines 603-607).

4. The authors did not measure any GLUT transporter, whether oxidized or reduced; GLUT transporter status is not shown.

Response : Since the goal of our study was to evaluate mitochondrial functions upon different glucose exposure, it did not seem relevant to us to study this aspect. However, the first experiences were conducted to check that the different glucose treatments had metabolic impact; by measuring lactate production under glucose treatments (S1 Fig), we can state indirectly that GLUTs were not limiting factors.

5. And neither is cell death, which means there was no cell death in the hypoglycemic phase. 

Response : We did not specifically measure cell death when using the different glucose treatments. As cell death can be easy detectable by microscopy since cells are floating in the culture medium, we can affirm that low glucose exposure either for 12 or 72h did not promote HL-1 cell death. However, when cultured for long time (>3 weeks) in normal glucose medium (1 g/l), we observed a decrease in cell proliferation compared to the high glucose medium.

6. How was mitochondria function affected to produce superoxide anion instead of water?

Response : This is a difficult question to answer; from our data and in our experimental conditions, we observed a higher superoxide anion production and protein oxidation upon high glucose (HG) and intermittent high glucose (IHG) compared to low and normal glucose conditions. We can hypothesize that when increasing electron flow through the respiratory chain, we increase the probability of electron leak to generate reactive oxygen species. This could be deleterious for mitochondrial respiratory chain as demonstrated by Zuo et al. (2009) after ischemia/reperfusion when ROS production is highly increased; furthermore, those mitochondrial alterations can be reversed by the addition of an antioxidant. However, we showed that only IHG treatment but not HG was able to impact the mitochondrial respiration indicating that superoxide anion production is probably not the only mechanism to promote mitochondrial dysfunction (Fig 4A, succinate).

7. The authors should perform WB for Glut-1 and 4 to see which one is involved and fluorescent microscopy to see how cells are affected morphologically.

At this point, the manuscript is very confusing and hard to follow.

Response : Since Glut-1 and 4 expression in HL-1 cells was previously studied (Chaudary et al., 2002; Shuralyova et al., 2004) and we used the same cells and cell culture conditions, we did not perform such experiments. We rather used indirect parameters (glucose consumption and lactate production) even if there are less sensitive than a direct determination of glucose uptake. Once again, the goal of our study was to examine mitochondria under different glucose exposure but not the metabolic flux.

Those references were added in the discussion.

Once again, we thank both reviewers for their useful comments and we appreciate your time reading our manuscript.

Looking forward to reading your response.

Yours sincerely,

Pr Stéphane Allouche

---

## [Decision Letter · Decision Letter 1]

20 Jul 2023

Glucose fluctuation promotes mitochondrial dysfunctions in the cardiomyocyte cell line HL-1

PONE-D-22-24271R1

Dear Dr. Allouche,

We’re pleased to inform you that your manuscript has been judged scientifically suitable for publication and will be formally accepted for publication once it meets all outstanding technical requirements.

Kind regards,

Kanhaiya Singh, Ph.D

Academic Editor

PLOS ONE

Additional Editor Comments (optional):

Reviewers' comments:

Reviewer's Responses to Questions

**Comments to the Author**

1. If the authors have adequately addressed your comments raised in a previous round of review and you feel that this manuscript is now acceptable for publication, you may indicate that here to bypass the “Comments to the Author” section, enter your conflict of interest statement in the “Confidential to Editor” section, and submit your "Accept" recommendation.

Reviewer #1: All comments have been addressed

Reviewer #2: All comments have been addressed

2. Is the manuscript technically sound, and do the data support the conclusions?

Reviewer #1: Partly

Reviewer #2: Yes

3. Has the statistical analysis been performed appropriately and rigorously? 

Reviewer #1: Yes

Reviewer #2: No

4. Have the authors made all data underlying the findings in their manuscript fully available?

Reviewer #1: Yes

Reviewer #2: Yes

5. Is the manuscript presented in an intelligible fashion and written in standard English?

Reviewer #1: Yes

Reviewer #2: Yes

6. Review Comments to the Author

Reviewer #1: The authors respond most of the critics in a logical manner. I would recommend to consider it for publication.

Reviewer #2: (No Response)

7. PLOS authors have the option to publish the peer review history of their article (what does this mean?). If published, this will include your full peer review and any attached files.

Reviewer #1: **Yes: **Anandhan Annadurai

Reviewer #2: No

---

## [Editor Report · Acceptance letter]

14 Sep 2023

PONE-D-22-24271R1 

Glucose fluctuation promotes mitochondrial dysfunctions in the cardiomyocyte cell line HL-1 

Dear Dr. Allouche:

I'm pleased to inform you that your manuscript has been deemed suitable for publication in PLOS ONE. Congratulations! Your manuscript is now with our production department. 

Kind regards, 

on behalf of

Dr. Kanhaiya Singh 

Academic Editor

PLOS ONE